# Ecological Strategies for Resource Use by Three Bromoviruses in Anthropic and Wild Plant Communities

**DOI:** 10.3390/v15081779

**Published:** 2023-08-21

**Authors:** Bisola Babalola, Aurora Fraile, Fernando García-Arenal, Michael McLeish

**Affiliations:** 1Centro de Biotecnología y Genómica de Plantas (CBGP), Universidad Politécnica de Madrid (UPM) and Instituto Nacional de Investigación y Tecnología Agraria y Alimentaria (CSIC/INIA) and E.T.S.I. Agronómica, Alimentaria y de Biosistemas, Campus de Montegancedo, UPM, 28223 Pozuelo de Alarcón, Madrid, Spain; 2School of Agriculture, Food and Wine, The University of Adelaide, Adelaide, SA 5005, Australia

**Keywords:** host range, transmission, metagenomics, emergence, habitat heterogeneity, co-occurrence

## Abstract

Ecological strategies for resource utilisation are important features of pathogens, yet have been overshadowed by stronger interest in genetic mechanisms underlying disease emergence. The purpose of this study is to ask whether host range and transmission traits translate into ecological strategies for host-species utilisation in a heterogeneous ecosystem, and whether host utilisation corresponds to genetic differentiation among three bromoviruses. We combine high-throughput sequencing and population genomics with analyses of species co-occurrence to unravel the ecological strategies of the viruses across four habitat types. The results show that the bromoviruses that were more closely related genetically did not share similar ecological strategies, but that the more distantly related pair did. Shared strategies included a broad host range and more frequent co-occurrences, which both were habitat-dependent. Each habitat thus presents as a barrier to gene flow, and each virus has an ecological strategy to navigate limitations to colonising non-natal habitats. Variation in ecological strategies could therefore hold the key to unlocking events that lead to emergence.

## 1. Introduction

Forecasting plant-virus disease risk is a necessity, as viruses cause the largest fraction of emerging diseases of plants [1]. Central to virus emergence are changes in the virus host range, that is, the number of host species used by a virus [2,3,4]. Host-range evolution is incompletely understood, as it depends on multiple factors, some intrinsic to the virus, such as genetic traits that determine its fitness in different hosts, and some extrinsic to the virus, related to ecological strategies (i.e., ecological and epidemiological traits) that determine host-plant resource utilisation (e.g., the potential for encountering hosts) across plant communities [5,6,7]. The degree of similarity in resource utilisation (niche overlap) by species, in respect to community structure [8], can be articulated by the co-occurrence of viruses in individual host plants across habitats of an ecosystem. Experimental analyses of intrinsic factors in plant-virus host-range evolution have focussed on understanding specificity of infection, which implies that the fitness of a virus varies across its potential hosts. These studies have underscored the relevance of across-host fitness trade-offs in host-range evolution that occur when adaptation to one host involves fitness penalties in another. Causes of across-host fitness trade-offs are pleiotropic effects of, and epistatic interactions among, host range mutations, on one side, and host factors on the other [9,10].

Ecological factors in host-range evolution are less amenable to experimentation (but see [11,12]), and their analysis in wild plant communities is still very limited, because viruses have been studied primarily as pathogens of crops [13,14,15]. Host-range evolution to a subset of potential hosts depends on transmission, species interactions, and heterogeneity in resources [16,17] and, ultimately, on the distribution and movement of viruses across complex communities [18,19]. Despite the use of high-throughput sequencing (HTS) to characterise viral communities in non-agricultural plant communities [20,21,22,23,24,25,26,27], virus host ranges across native plant assemblages, and the relationship between host range and virus genetic diversity, remain opaque at best [6], which are major limitations to predicting virus emergence.

To contribute to an understanding of the ecology of host-range evolution, here we analyse by HTS of plant RNA the host resource utilisation and the genetic diversity of three viruses, cucumber mosaic virus (CMV), tomato aspermy virus (TAV) and pelargonium zonate spot virus (PZSV), that have broad host ranges and occur in heterogeneous agricultural ecosystems in central Spain [28]. These three virus species belong to the genus *Cucumovirus* (CMV and TAV) or *Anulavirus* (PZSV) of the family *Bromoviridae*, and as such they share the structure of virus particles and genome, and similar strategies for gene expression [29]. Briefly, the isometric particles encapsidate a tripartite, single-stranded, messenger-sense RNA genome, with segments named RNA1, RNA2, and RNA3 according to decreasing size. RNA1 encodes the ~110 kDa protein *1a*, with methyltransferase and helicase motives, RNA2 encodes the 79–85 kDa protein *2b*, which is the RNA-dependent RNA polymerase, and RNA3 encodes the 34 kDa movement protein (*MP*) and the 22–24 kDa coat protein (*CP*), which is translated from subgenomic RNA4. The 5′ end of the genomic segments has a methylated cap structure, and the 3′ end has a complex structure conserved among the three genomic RNAs. In the cucumoviruses, but not in PZSV, this structure mimics a tRNA and can be amino acylated. A major difference between both genera is that cucumoviruses, but not anulaviruses, encode a second protein in RNA2, the ~19 kDa protein *2b*, which is a silencing suppressor protein [30].

Despite these common features, the very uneven current information on these viruses suggests that CMV, TAV, and PZSV differ in traits that may condition host resource utilisation in plant communities. CMV is by far the most studied of the three viruses, both regarding its molecular biology and its ecology and evolution (reviewed in [31]). CMV is an important pathogen of many horticultural and field crops all over the world, and is the plant virus with the broadest reported host range, including 1071 species in 100 families of mono- and dicotyledon plants at the time of the last review [32]. Many isolates of CMV from different host-plant species have been characterised, showing a large diversity regarding the nucleotide sequence of their genomes, the host range, and the pathogenicity and virulence in different hosts. In terms of similarity of the nucleotide sequence of the genome, CMV isolates are clustered into Subgroup I and Subgroup II, which are approximately 70–78% similar according to the *1a*, *2a*, *MP* or *CP* genes, with group similarity being greater than 88% [30]. In central Spain, only Subgroup I isolates have been detected [33]. All isolates of CMV are transmitted horizontally by aphids in a non-persistent manner. More than 80 species of aphids have been shown to transmit CMV, and specificity in transmission efficiency on the part of the virus is determined solely by the *CP* [34]. CMV has been shown to be transmitted vertically through the seed in more than 40 host species, with extremely variable rates (0.5–100%) according to virus isolate and plant species and genotype [35]. TAV has been reported to cause a relevant disease in chrysanthemums and has been less studied than CMV, with which it shares the mechanisms of horizontal and vertical transmission. Information on the natural host range is very limited, and the experimental host range includes 100 species, mostly in the Asteraceae, Chenopodiaceae, and Solanaceae [36]. Genetic diversity of characterised TAV isolates is low [30]. PZSV was first isolated from *Pelargonium zonale* plants and, shortly after, was reported as the cause of an important disease of tomatoes in Southern Italy [37] and, about twenty years later, in other countries of Europe, the US, and Israel [38]. It has also been reported as a pathogen of peppers, artichokes and sunflowers in different regions of the world [38,39,40,41]. In Australia, it has been reported as infecting wild plants, the self-introduced *Cakile maritima* and the indigenous *Anthoceris ilicifolia* [42,43], and in Japan it infects riverine communities of wild Brassicaceae in central Honshu [24]. In contrast to CMV and TAV, PZSV is not aphid-transmitted, has no known vector and is not transmitted through the soil [37,38]. Horizontal transmission is through the pollen, and may be facilitated by pollen-carrying flower thrips that mechanically injure leaves or flowers [44,45]. Seed transmission of PZSV, possibly through the maternal tissues and through the pollen, has been shown in tomatoes, the experimental host *Nicotiana glutinosa*, and the natural wild host *Diplotaxis erucoides* [37,38]. The few characterised isolates show high sequence identity of the genome sequences, but differ in experimental host range, and in their capacity of seed transmission in different hosts [38].

In this work we characterise the host range, the genetic structure, and the diversity of CMV, TAV, and PZSV in plant communities of a heterogeneous ecosystem in central Spain that occupy habitats with different levels of human intervention. With these data we test the hypothesis that the different host range and transmission traits of these three viruses translate into different ecological strategies for host-plant resource utilisation across plant communities. Results show that the three viruses have strong habitat-specificity in host resource utilisation. The ecological strategies associated with habitat specificity differ in the way host species are utilised by the three viruses. Variation in host use within virus species also depends on habitat type.

## 2. Materials and Methods

### 2.1. Description of Study Sites and Sampling

We conducted 78 collections at 23 sampling sites between July 2015 and June 2017 in the Vega del Tajo-Tajuña agricultural region of south-central Spain (Figure 1). These collections produced 6291 individual samples of 271 plant species distributed over four habitats, each with distinct cover types. The first two of the four habitats constitute what we term the ‘anthropic habitats.’ Crops (Crop) are annual monocultures and may include assemblages of wild plant species. Crops are left fallow between seasons or rotated. Narrow boundaries that separate crops (Edge) are relatively permanent communities of wild species that partially benefit from nutrient and water supplementation in adjacent crops. The second two habitat categories, which we term ‘non-anthropic’, are firstly evergreen oak forests (Oak) that are the region’s primary habitat for wild species. Oak is dominated by *Quercus* species with an understory of sclerophyllous shrubs and grasses. Structurally distinct from Oak are abandoned, undisturbed and patchy areas of successional scrubland (Wasteland) interspersed among the other habitats. The terms collection, site, and habitat represent three levels of experimental unit. Four sites each of Oak and Wasteland, with individual collection produced from the re-sampling of sites of these habitats, were visited in autumn and spring over two growing seasons. Collections from Edge and Crop, with four and eleven sites, respectively, were produced from visits during spring, summer, and autumn [46]. Plant species with an abundance of 5 or more individuals in at least one of the four habitats were retained for HTS, as these are central to transmission among habitats. Abundant species that occurred in a single habitat were also retained. The sample of 271 plant taxa was reduced to 118 species for HTS.

### 2.2. Detection of Viral Reads and Estimation of Host Range

Individual RNA extracts of leaf samples from the same site, collection, and plant species were pooled to obtain a single library preparation. In total, 323 libraries of 2037 pooled extracts were sent for sequencing (for details see [28]). Paired-end reads of 125 or 150 nt. were sequenced on Illumina HiSeq platforms. All reads were provided with Phred quality scores greater than Q30 and trimming of adapter contamination conducted with *cutadapt* v1.8.3 [47]. Genomic references of ssRNA viruses were accessed from NCBI’s Viral Genome Browser (https://www.ncbi.nlm.nih.gov/genomes/, accessed on 1 December 2018) and used to construct a local BLAST database. Blast+ version 2.2.29 [48] was used to identify virus OTUs derived from the HTS libraries. The results of the BLAST queries of each library were merged with taxonomic and study site details.

The detection of an operational taxonomic unit (OTU) was considered credible if a read met the following criteria: (1) having a BLAST query coverage of 100%; (2) a query length of 125 nucleotides (together referred to as HTS2C detections). Validation of the BLAST credibility criteria for HTS detections was undertaken with RT-PCR. Primers were designed from Coat Protein (*CP*) gene region references of CMV Subgroup I, TAV, and PZSV with the NCBI Primer BLAST tool (Table A1) for detection of these viruses in a subset of 208 randomly chosen libraries that represented 69 plant species from all four habitats. The specificity of primers was verified by Sanger sequencing of the amplicons from 3 to 5 RT-PCR-positive libraries of different plant species. Additionally, RT-PCR primers were assessed for specificity with five libraries from five plant species that were unlikely hosts of CMV, TAV, or PZSV, that is, libraries absent from the list of HTS2C detections, with no amplification in any case. Combined HTS and RT-PCR protocols were used to estimate host range, defined as the number of host species used by a pathogen.

### 2.3. Virus Set Memberships and Co-Occurrence Simulations

As there was more than one HTS library prepared for a single host species, the aggregation of counts from multiple libraries of the same host species may mistakenly conflate single infections into co-occurrences. Virus species set memberships (Figure 2), implemented with the R [49] package *eulerr* [50], were conducted by scoring the presence or absence of each of the three viruses either in each single host species or HTS library.

The C-score index (“checkerboard score”) was used to compare virus species co-occurrence patterns [51] among populations defined by habitat, season, or site. The larger the C-score, the fewer shared sites (i.e., host species) there are. The matrix-wide average can contain individual pairs of species that are segregated, random, or aggregated. Two simulation algorithms that differ in how host species are treated were used to test whether these patterns differed from random occurrences across hosts, libraries, seasons, and sites. Absence–presence matrices treated host species as either equiprobable (Sim2), or where the probabilities of occurrence at sites is proportional to the observed virus richness at the site (Sim4). Viruses are assumed to colonise host species randomly with respect to one another. The two algorithms were selected and compared as they have the lowest error frequencies. The co-occurrence simulations were implemented with the R package *EcoSimR* [52].

### 2.4. Genetic Diversity and Population Genomic Analysis

Reads that satisfied the HTS2C credibility criteria were aligned with their respective reference genome using the Burrows–Wheeler Alignment (BWA) algorithm [53]. Variant calling analysis was conducted with the binary alignment map (bam) files generated from the BWA output, and implemented with BCFtools [54]. The genetic diversity analyses and population genomics were performed with the R package *PopGenome* [55].

A subset of read sequence libraries with alignments that produced a sufficient depth (Table A3) were used for population genomic analyses. As whole genomes may not be recovered at even depths [56], libraries with at least 10× coverage of the genome were included in the analyses. Pilot analyses of genetic diversity of CMV, in which libraries with progressively lower coverage were included, indicated that libraries with as low as 10× coverage depth could be used for estimating genetic diversity without introducing errors or bias. Genetic diversity of CMV at the site and habitat levels was estimated with 38 libraries from four habitats, while information from 28, 20 and 39 Edge libraries for CMV, TAV and PZSV, respectively, was used for analyses at site level. The nucleotide diversity (*π*), that is, the average pairwise difference between all possible pairs of individuals in a sample, was used as a measure of genetic diversity (*π* = *S*/*L* where *S* is the number of sites with more than one nucleotide variant and *L* is the number of nucleotides in the sequence). Whole-genome consensus sequences (WGCS) were used to estimate within-population genetic diversities using the Tamura 3-parameter model [57], with standard errors of each estimate based on 1000 bootstrap replicates, as implemented in MegaX [58]. Missing data was allowed when it accounted for not more than 30% of the sequence. Populations were defined either by habitat or study site.

A sliding-window analysis was used to calculate nucleotide diversity along the genome (i.e., averaging across several variants). The approach allows for a systematic examination of localised nucleotide diversity across the genome. As the genomes of CMV, TAV, PZSV are tripartite, *π* was estimated over the full length of each segment. A window size of 1000 nt was used for all viruses, with a shift of 5 nt along the genome made for each estimate of *π.* Heatmaps were implemented with the R package adegenet v2.1.5 [59] and used to visualise single nucleotide polymorphisms (SNPs) of genotypes in respect to the respective reference genomes of each virus.

### 2.5. Population Genetic Structure

To investigate the strength of association between CMV genotypes and habitat, level of anthropic disturbance (i.e., Crop with Edge and Oak with Wasteland), site, or host species observations (i.e., traits), phylogenetic inference and Bayesian tip association significance (BaTS) tests [60] were conducted. The BaTS analysis was used to test the observations inferred from the phylogenies, and generated an association index (AI) and parsimony score (PS). The AI index measures the overall strength of association between genotypes at tips of phylogenetic trees and traits (the imbalance of internal nodes of the phylogeny), and the PS index measures the degree of homoplasy between genotypes in the phylogenetic tree and tip values. Thus, low AI values represent strong phylogeny–trait association and the PS value is inversely related to the strength of tip-character association [60,61]. The BaTS approach repeatedly simulates associations under the null hypothesis that characters at the tips are randomly distributed across the phylogeny. The resulting *p*-values follow a uniform distribution, and the type 1 error of the test will be correct for all levels of statistical significance. Significant AI and PS *p*-values indicate that the observed association and level of homoplasy between genotypes and traits is unlikely to have occurred by chance alone. Whole-genome consensus sequences were aligned with the CLUSTALW algorithm [62] implemented in MegaX. The General Time Reversible model (GTR) with gamma distributed substitution rates and invariant sites allowed was implemented in Mr. Bayes. Posterior probabilities and mean branch lengths of Bayesian consensus phylogenies were derived from 30,000 post-burnin trees. A random sample of 2000 posterior post-burnin trees from a Bayesian inference implemented with MrBayes v3.2.7 [63], were used to account for phylogenetic uncertainty in the tip-association analysis. Four Markov chains were run for 20 million generations, sampling each chain every 500 trees. Convergence and posterior parameter distributions were assessed using the MCMC Tracer Analysis Tool v1.7.1.

## 3. Results

### 3.1. Detection of Viral Reads and Estimation of Host Range

The detections of OTUs by HTS2C and RT-PCR approaches are summarised in Table 1 according to each host species and the number of libraries pooled per host. The random subset of libraries selected for RT-PCR accounted for 90% of the host species used for HTS2C detections. Correspondence between the host-range estimates by habitat derived from each approach was significant (*χ*^2^ = 1.948, d.f. = 11, *p*-value = 0.999; and Kendall’s rank correlation *z* = 4.3506, tau = 0.977, *p*-value < 0.0001) (Table A2), confirming the accuracy of HTS2C detections. Thus, for subsequent ecological analyses, host range was derived from the HTS2C counts. The detections by the HTS2C approach resulted in host-range estimates of CMV, TAV and PZSV as 88, 38, and 67, respectively, at the ecosystem level. Mean percentage identity of local BLAST queries between reads and reference genomes of the three viruses under the HTS2C criteria were between 92.7% to 98.9% (Table A4).

The observed host range of TAV (*n* = 38) was half that of PZSV (*n* = 67) and was broadest in CMV (*n* = 88). The three viruses utilised host species mostly in Edge that had the highest host-species richness among the habitats (Table 1 and Figure A1). Host-species resource utilisation by each of the viruses differed with greater similarity observed between CMV and PZSV (Figure A2). The realised host range of the three viruses (Table 1 and Figure 2) in each of the habitats was lowest for TAV in Wasteland, with the highest observed for PZSV in Edge. The broadest realised host range in all habitats apart from Edge was observed in CMV. The relatively narrow host range of TAV goes some way to explaining the rarity of this virus in the ecosystem.

### 3.2. Virus Set Memberships and Co-Occurrence Simulations

Independent of the host-range breadth is the co-occurrence or specificity of the viruses in respect to particular host species. As RNA extracts of individual samples of host species were pooled to produce HTS libraries, infection may have occurred in any one of the individual samples that made up the pool. Figure 2 shows counts of each virus in terms of observations made either in a host species or HTS library. Although it is likely that some counts of a species did not include true cases of co-occurrence in an individual library, the counts of set memberships produced similar distributions. Several patterns are evident: (1) infections either by CMV or PZSV in the absence of any of the other two viruses occurred at a higher frequency than for TAV; (2) the number of infections either by CMV or PZSV, in the absence of the other two viruses, varied between habitats; and (3) multiple infections of any combination of all three viruses occurred at the highest frequency in Edge. Furthermore, the number of observations of multiple infection tended to be similar across the sites of Edge (Figure A3, top), and indicated that counts (i.e., spatial variation) in any category of the set memberships were relatively consistent. The exception was the variation in the co-occurrence of both CMV and PZSV between site L1 and the other three sites. Seasonal (i.e., temporal) variation in the occurrence of the viruses in Edge (Figure A3, bottom) was substantial especially in the high proportion of infections by all three viruses that occurred in spring. The pattern of seasonal variation in virus occurrence differed according to the habitat (Figure A3 (bottom) and Figure A4), with CMV and PZSV mostly occurring in single infections during the spring in Oak and Wasteland, respectively. Also, autumn infections of CMV were most frequent in Edge, in co-occurrence with PZSV, and in Wasteland, in single infections.

Simulations in co-occurrence were run to test whether the set-membership observations agreed with differences in aggregation and segregation of viruses among host species of each habitat. The C-score for Edge was significantly lower (*p-*value = 0.017) than expected (Table 2), which indicated aggregation of virus species pairs in a proportion of host species of this habitat. This result agreed with the high proportion of multiple infection observed in Edge (Figure 2). Similarly, each of the four Edge sites had significantly lower (*p-*value < 0.001) than expected C-scores (not shown). The high frequency of multiple infections by all three viruses in Edge shown by the set memberships, indicated that the viruses co-occur in a proportion of host species in this habitat. The higher-than-expected C-score of Oak and Wasteland indicates a significant proportion (*p-*value ≤ 0.001) of virus species pairs are segregated and tend not to co-occur in host species of the non-anthropic habitats. Together, each of the three viruses exhibited distinct patterns of occurrence across the habitats. PZSV occurred in host species in Edge without the presence of the other viruses that tended to co-occur in this habitat. Similar host utilisation by CMV and PZSV (Figure A2) was a result of the low abundance (or titre) of TAV across all habitats. TAV was only present in Oak without co-occurring with either of the other viruses. Lastly, host communities in Edge by far supported the highest frequency of multiple infection.

### 3.3. Genetic Diversity Analysis

Variant calls were used to infer SNPs and assess nucleotide diversity for each virus genomic segment and population. Mean *π* at the site level (Table A5) was between 0 and 0.007 for all segments and viruses except for RNA2 of PZSV at mean *π =* 0.01, and indicated consistent low genetic diversity among sites. At the habitat level, CMV had higher *π* than at the site level, particularly in RNA3 in Crop and Edge habitats (Table 3). A sliding-window analysis at the site level of localised nucleotide diversity (*π*) across segments of CMV (Figure 3 and Figure A5), TAV (Figure A6), and PZSV (Figure A7), indicated the highest diversity was observed in the RNA3 segment of CMV and the lowest in the RNA1 and RNA 3 segments of PZSV. Both CMV and TAV had higher nucleotide diversity in their RNA3 genome compared to the RNA1 and RNA2 segments. For CMV, nucleotide diversity trended in similar ways across each segment for each habitat. A Kruskal–Wallis test showed that ranks of *π* did not differ significantly across the RNA segments. The nucleotide diversity trends of TAV and PZSV across the segments for Edge sites was not as clear, particularly for RNA1 and RNA2. By comparison to that of CMV, the very low values of *π* in these instances indicate that variation was driven by mutations at a very few sites. Together, similar trends of localised variation in *π* across the genomic segments, in instances where nucleotide variation was not minimal, shows that ecosystem- and site-level factors do not correspond to variation in genetic diversity. The observed variation in *π* between habitats or sites of each virus may not be independent of sample size.

### 3.4. Population Genetic Structure

Whole-genome consensus sequences from 38 libraries of CMV were used to infer phylogenetic relationships among virus genotypes and their habitat affiliations as the reads of TAV and PZSV could only be assembled for Edge sites. The general patterns of branching (Figure 4) are consistent with the clustering of genotypes by habitat, with most major clades including multiple host species from taxonomically distant families. In all three RNAs, isolates from Oak cluster separately from those of Edge plus Crop. The strong support for branching along the backbone of the phylogeny of each genomic segment indicates Crop and Edge (and instances of Wasteland) isolates that evolved from a variant that occurred in Oak. The next most-derived clades are isolates that largely occurred in Crop and Edge, implying these variants re-colonised Crop (and Oak and Wasteland) from host species of Edge. A very long branch in the RNA3 segment indicated a relatively divergent clade from Edge and Crop, which were split between *Convolvulus arvensis* and *Cucumis melo,* respectively. Overall, the sliding-windows analysis and phylogenetic reconstructions show similar evolutionary dynamics. The variation in the genetic diversity across the genomes of all RNA segments and for each RNA between habitats is indicative either of fluctuations in the abundance of each segment or of sample size. Although isolates from Wasteland were rare, in each segment there was phylogenetic over-dispersion of isolates from this habitat, which was consistent with colonisation cycling among the other three habitats. Overall, the inferences are consistent with habitat-specificity more so than host-species specificity in host resource utilisation by CMV.

Bayesian tip-association significance testing for all three RNAs of CMV showed significantly (*p* < 0.0001) strong (Table 4) associations that indicated a strong effect of anthropic disturbance, or habitat, on genetic diversity. The strong tip associations indicated by the AI statistic showed a high degree of coherence among isolates sampled from the same habitat, which were descended from very few ancestral genotypes.

## 4. Discussion

Understanding the ecological strategies that modulate host range is central to forecasting disease risk and virus emergence [4,64]. Most metagenomic studies either explore the richness of viromes or do not address explicitly plant-virus ecological strategies [20,21,22,23,24,25,26,27]. We have focussed on the interactions of multiple plant viruses and higher-order interactions among them in respect to heterogeneity in the resources available from host-plant communities.

Here we analyse the ecological strategies for resource use of three viruses, CMV, TAV and PZSV, that share features of particle and genome structure and gene expression, as they belong to the same family, *Bromoviridae* [29]. Common features of their single-stranded, messenger sense, tripartite RNA genomes are encoding in the two larger genome segments, RNA1 and RNA2, proteins 1a and 2a, respectively, which are involved in genome replication and are part of the viral replicase [65]. Dicistronic RNA3 encodes the MP, with the function of gating plasmodesmata, and necessary for cell-to-cell and systemic movement within the infected plant [66], and the *CP*, also required for systemic movement and for virus dispersal as it forms the virus particles [29,66]. Despite similarities in genomic structure and gene function, CMV, TAV and PZSV differ in traits relative to host range and mode of transmission, which we hypothesise will translate into different ecological strategies for host-plant resource utilisation across plant communities.

Our study is based on virus detection by HTS of rRNA-depleted plant RNA in 323 libraries of 118 plant species from communities growing in four different habitats, more (Crop and Edge) or less (Oak and Wasteland) anthropic, in a heterogeneous ecosystem in central Spain. As HTS detection of virus OTUs involves uncertainty from different factors that include the sensitivity of the detection method, the proportion of false positives produced, or the high proportion of unknown virus species expected in wild plant communities [23,67], detection was confirmed by RT-PCR with virus-specific primers on a random subset of 208 libraries from 69 plant species. The very good correspondence between RT-PCR and HTS detection validated HTS2C detection, which was the basis for further analyses.

According to our hypothesis, we show that differences in ecological traits translate into variation in how host resources are used. Thus, CMV and PZSV, which are less related taxonomically as they belong to different genera and are transmitted by different mechanisms, share a more similar pattern of host-species utilisation than CMV and TAV (Table 1, Figure A2), which belong to the same genus, are both vectored by aphids, and are so closely related genetically as to be able to form stable hybrids by reassortment of genomic segments [30]. The three viruses had their broadest realised host range at Edge, but while TAV and PZSV are mainly Edge viruses, with about 75% of their hosts in this habitat, only about half of the hosts of CMV were from Edge, with a large fraction of hosts from the less anthropic habitats, Oak and Wasteland (Table 1, Figure 2). The patterns of virus detection in single occurrence and in co-occurrence over space (habitats) and time (seasons) (Figure 2, Figure A3 and Figure A4) show that the three viruses have different ecological strategies that are habitat-dependent. While in Edge CMV and TAV co-occur with PZSV in most hosts, PZSV has a set of non-shared hosts in this habitat and hosts unique for CMV and TAV occur mostly in the less anthropic habitats. The relatively consistent maintenance of the viruses across space, shown across sites of Edge, contrasts with the temporal variation through the seasons. While most PZSV detections occur in spring in all habitats, most TAV detections in all habitats are from autumn. CMV detections are most frequent in spring in Edge and Oak, and in autumn in Wasteland. Maintenance of CMV and PZSV in the autumn is mostly in Edge co-occurrences or, for CMV, as single occurrences in Wasteland. This underscores that infection of host plants and host plants of specific habitats by the three viruses is dynamic.

Despite the different function the four habitats have in the ecology of these viruses, Edge stands out as a reservoir community with a high virus-host richness, where virus–host interactions are mostly organised by interactions among the three viruses in shared hosts. These features of Edge agree with our previous results from the analyses of different plant–virus interaction data sets [7,28]. Edges have features that may explain these results. As Edges benefit from water and nutrient supplied to nearby crops, they provide plant communities with a more stable environment over the year than the wild habitats, where plant growth and reproduction ceases during winter and summer. As a consequence, the temporal variation of plant cover and biomass over the year is less than in the other habitats [68]. However, the role of Edge as a reservoir was not apparent in an analysis of patterns of infection of four contact-transmitted tobamoviruses in the same ecosystem using the same set of HTS libraries [67], as detections were not more numerous at Edge than at the less anthropic habitats. Thus, the reservoir role of Edge might also be explained, at least in part, by its role in maintaining insect populations, which has been repeatedly shown for pollinators and for predators and parasitoids of agricultural pests, and could well be so for insect vectors of viruses [69,70,71]. Lastly, Edge may be a sink for aphid populations, as the first stimulus for aphid alighting in plants is visual cues in the 500–600 nm wave length (yellow-green) that are best perceived in contrast with other colours, such as the brown colour of bare soil [72,73]. The patterns of virus co-occurrence are not explained solely on the basis of virus-host ranges (TAV mostly occurs in infections with the other viruses, and a majority of all co-occurrences involve PZSV rather than CMV). This observation is not explained by shared mechanisms of virus transmission as most co-occurrence involves PZSV, which is not aphid-transmitted. This suggests higher order interactions are a factor in host-species utilisation. It is tempting to speculate on the role of virus infections on the behaviour of aphids and other insects, and on the growth of their populations. As for other aphid-transmitted viruses, infection by CMV has been shown to alter plant volatile composition in different host species, making infected plants more attractive to aphids and to pollinators, also altering the behaviour of insect herbivores, predators, and parasitoids [74,75,76,77,78,79]. Thus, plants initially infected by CMV, the virus with the broadest host range, could attract aphids that transmit TAV, and thrips and other pollinators that would carry pollen from PZSV-infected plants. Of course, we cannot assume that other viruses outside the attention of this study do not have a part in the observed patterns of infection.

The dissimilarity in ecological strategies across habitats is not congruent with either the low genetic variation of each virus in the different habitats, nor the low genetic variation of CMV across segments and habitats. Although there was clearly localised variation in regions of the segments that differed between them, the trends along the segments were relatively consistent among habitats. The exceptions were the relatively high nucleotide diversity of RNA3 of CMV in Crop and Wasteland (Table 3), the two habitats where CMV isolates appear phylogenetically over-dispersed at RNA 3 (Figure 4). RNA3 encodes proteins related to host colonisation and transmission that, a priori, could show more variation across hosts than proteins involved in the replication of the RNA genome. However, low genetic diversity of the three viruses regardless of their host range, and the limited role of host species, compared to habitat, in shaping the genetic structure of the CMV population, argue against a major role of host specificity, and across-host fitness trade-offs, in the observed host ranges and patterns of infection.

For CMV, data allowed a more detailed analysis of population structure. The sample size of isolates used to estimate *π* was larger for Edge than for Oak or Crop, in that order, yet the variation in *π* of CMV in Edge shown by the sliding-window analysis was flat compared to that observed for the other habitats (Figure 3). The low genetic diversity of isolates infecting the melon crop and perennial or annual host species that grow along the year in Edge is consistent with early epidemiological analyses carried in the south of France and NE USA (reviewed in [80]) and more recently in central Spain [68] that indicated that wild plants growing in proximity to crops assure the year-round presence of the virus for crop infection. That Wasteland had a large variation may be the result of a small sample with extreme pair-wise differences compared to the Edge sample. This assertion is supported by the phylogenetic inferences (Figure 4) with isolates from Wasteland being phylogenetically over-dispersed within clades that predominate in one of the other habitats. This phylogenetic pattern is consistent with sporadic and successful colonisation from other habitats associated with population bottlenecks, which would counter virus adaptation to host or environment due to genetic drift [81]. Closely related isolates from Oak tend to be basal in the phylogenies of the three RNAs. This is indicative of genotypes from Oak successfully colonising all the other habitats, which occasionally re-colonise Oak.

## 5. Conclusions

In summary, our study has shown that genetically closely related viruses do not have to share similar ecological strategies for resource utilisation. The two less-related viruses, CMV and PZSV, had similar strategies by comparison with the closely related pair, CMV–TAV, and this strategy included a broad host range. A broad host range permitted wider resource-use opportunities among host species, but also a higher frequency of co-occurrence with the two other viruses in the host species they had in common, and this was habitat dependent; most co-occurrences were observed in the anthropic habitats. Each habitat thus presents as a barrier to gene flow, and each virus species has an ecological strategy to navigate host-species resource heterogeneity. The variation in ecological strategies, or rate at which they evolve, exhibited by a virus should correlate with the dimensions of its reservoir (niche) available across an ecosystem. The ecological strategies (e.g., phenotypic plasticity [82]) of each virus species will determine the potential for ecological fitting [83] and in overcoming barriers to colonisation of a non-natal habitat. Identification of variation in ecological strategies may therefore hold the key to predicting emergence events.

## Figures and Tables

**Figure 1 viruses-15-01779-f001:**
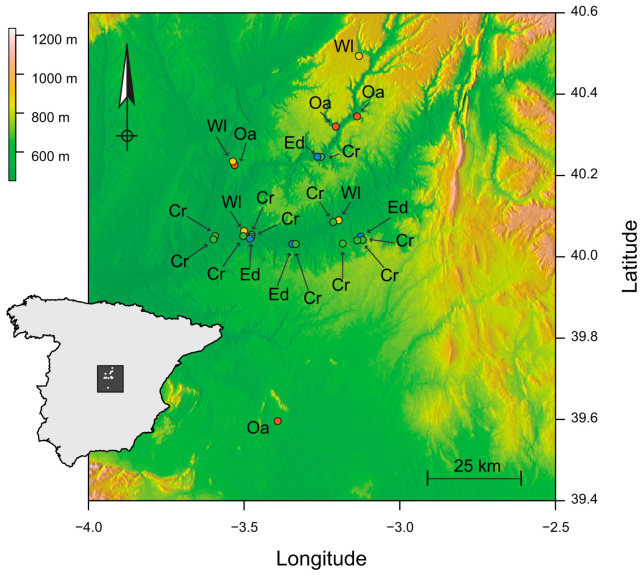
Study extent showing replicate sites of Crop (Cr, green), Edge (Ed, blue), Oak (Oa, red), and Wasteland (Wl, yellow) sites in central Spain (inset). Elevation indicated at top left.

**Figure 2 viruses-15-01779-f002:**
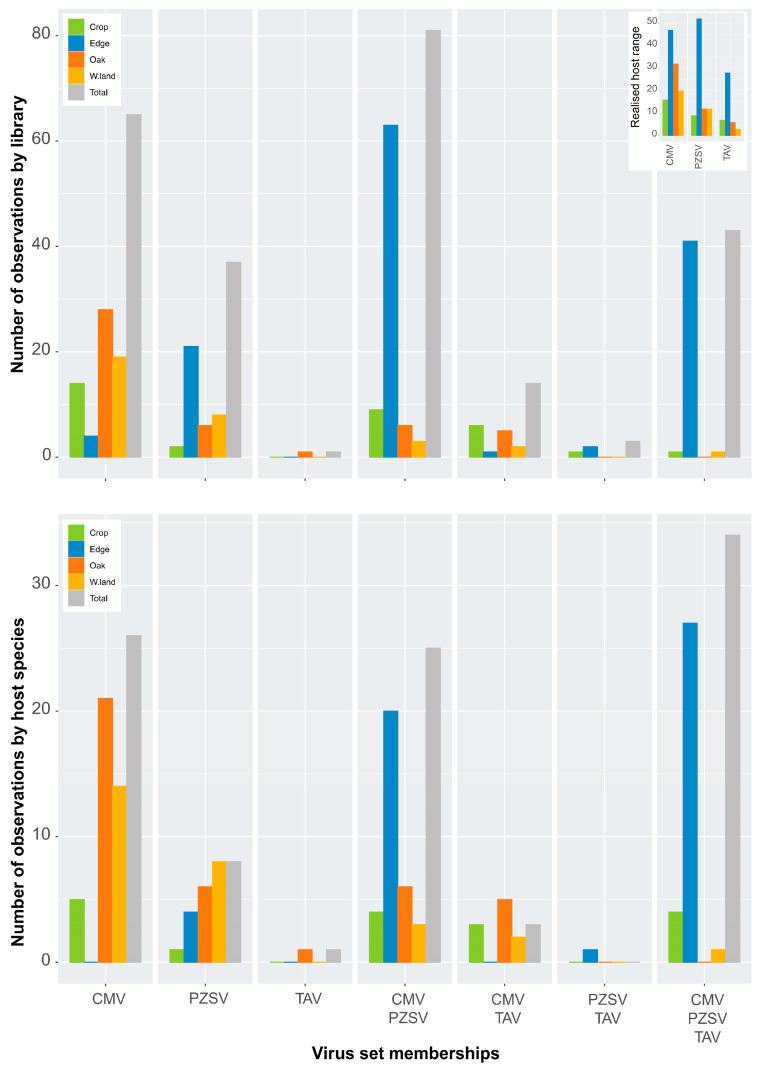
Set memberships (HTS2C) for virus occurrence in the habitats. The observations have been scored by counting from HTS libraries (above) and from host species (below). CMV, PZSV and TAV indicate single infections by each of these viruses. Virus combinations in multiple infections are also indicated. Upper right inset shows realised host range.

**Figure 3 viruses-15-01779-f003:**
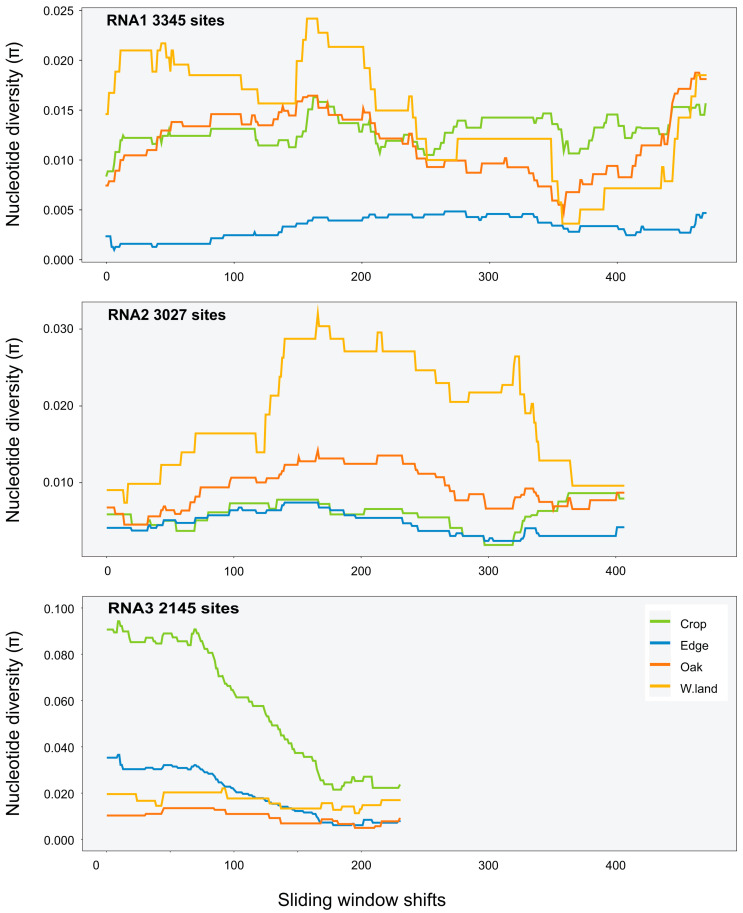
Sliding-windows analysis of nucleotide diversity (*π*) across the consensus genomes of CMV segments at the habitat level (sliding-window size was 1000 nt).

**Figure 4 viruses-15-01779-f004:**
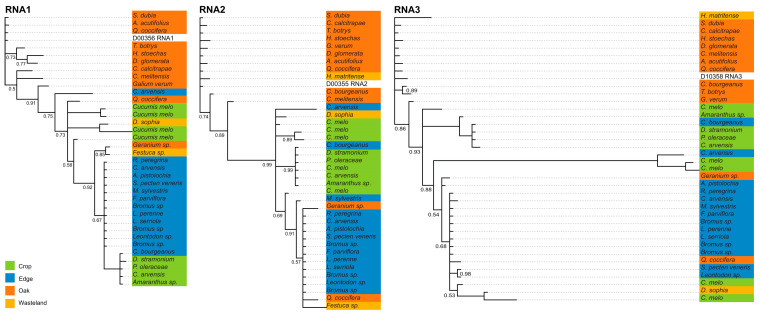
Bayesian maximum-credibility consensus phylogenetic inferences of the three genomic segments of CMV showing branches with 0.5 or higher posterior probability.

**Table 1 viruses-15-01779-t001:** Observed host range of three *Bromoviridae* virus species by two detection protocols. The OTUs were detected by high-throughput sequencing (HTS) and RT-PCR.

		CMV				TAV				PZSV			
Host Species	No. of Libraries (HTS2C)	CR	ED	OA	WL	CR	ED	OA	WL	CR	ED	OA	WL
*Allium sativum*	1	**+**									**+**		
*Amaranthus* sp.	4	**+**	**+**			**+**	**+**				**+**		
*Anacyclus clavatus*	3	**+**	**+**				**+**				**+**		**+**
*Anchusa undulata*	3		**+**		**+**						**+**		
*Anthriscus caucalis*	2		**+**				**+**				**+**		
*Aristolochia pistolochia*	2		**+**				**+**				**+**	**+**	
*Asparagus acutifolius*	3			**+**	**+**							**+**	**+**
*Asphodelus aestivus*	1			**+**									
*Asteriscus aquaticus*	2											**+**	**+**
*Astragalus incanus*	1			**+**									
*Avenula bromoides*	1			**+**									
*Bassia scoparia*	1										**+**		
*Brachypodium phoenicoides*	2		**+**				**+**				**+**		
*Brachypodium retusum*	3		**+**								**+**		
*Bromus* sp.	6	**+**	**+**	**+**	**+**		**+**		**+**	**+**	**+**		
*Carduus bourgeanus*	8	**+**	**+**	**+**			**+**	**+**		**+**	**+**		
*Centaurea melitensis*	1			**+**				**+**					
*Centranthus calcitrapae*	1			**+**				**+**					
*Chenopodium album*	5		**+**				**+**				**+**		
*Cirsium arvense*	1		**+**								**+**		
*Convolvulus arvensis*	14	**+**	**+**		**+**	**+**	**+**			**+**	**+**		
*Conyza bonariensis*	2		**+**				**+**				**+**		
*Conyza canadensis*	3		**+**								**+**		
*Cucumis melo*	8	**+**				**+**				**+**			
*Cynodon dactylon*	4		**+**				**+**				**+**		
*Cyperus longus*	1	**+**											
*Dactylis glomerata*	1			**+**									
*Datura stramonium*	3	**+**	**+**			**+**					**+**		
*Daucus* sp.	3		**+**								**+**		
Unknown 4	3	**+**	**+**							**+**	**+**		
Unknown 5	1												**+**
*Descurainia sophia*	2		**+**		**+**				**+**		**+**		
*Diplotaxis erucoides*	11	**+**	**+**		**+**		**+**			**+**	**+**		
*Diplotaxis* sp.	1		**+**				**+**				**+**		
*Diplotaxis virgata*	1			**+**									
*Echium vulgare*	2		**+**		**+**						**+**		
*Erodium cicutarium*	3		**+**	**+**							**+**	**+**	**+**
*Eruca vesicaria*	1		**+**								**+**		
*Eryngium campestre*	1											**+**	
*Festuca* sp.	1				**+**								
*Fumaria parviflora*	3		**+**				**+**				**+**		
*Galium verum*	4		**+**	**+**	**+**			**+**			**+**		**+**
*Geranium* sp.	4		**+**	**+**	**+**						**+**		
*Helichrysum stoechas*	1			**+**									
*Hieracium pilosella*	1				**+**								
*Hirschfeldia incana*	1		**+**				**+**				**+**		
*Hordeum matritense*	2				**+**				**+**			**+**	**+**
*Hordeum vulgare*	2	**+**				**+**				**+**			
*Hypericum pubescens*	1			**+**									
*Jasminum fruticans*	1			**+**									
*Klasea pinnatifida*	1									**+**			
*Lactuca serriola*	6		**+**				**+**				**+**		
*Leontodon* sp.	4		**+**	**+**	**+**		**+**	**+**			**+**		
*Lepidium draba*	4		**+**				**+**				**+**		
*Lithospermum arvense*	1										**+**		
*Lolium perenne*	4		**+**				**+**				**+**		
*Lotus corniculatus*	1			**+**									
*Malva sylvestris*	3		**+**				**+**				**+**		
*Marrubium vulgare*	2			**+**									
*Medicago* sp.	1				**+**								**+**
*Milium vernale*	2		**+**								**+**		
*Origanum vulgare*	1							**+**					
*Papaver rhoeas*	4		**+**		**+**		**+**				**+**	**+**	
*Phalaris minor*	2	**+**					**+**				**+**		
*Phlomis lychnitis*	1			**+**									
*Picris echioides*	8	**+**	**+**		**+**	**+**	**+**			**+**	**+**		
Poaceae	1		**+**								**+**		
*Portulaca oleraceae*	1	**+**				**+**							
*Potentilla* sp.	1		**+**				**+**				**+**		
*Quercus coccifera*	4			**+**								**+**	
*Quercus ilex*	2			**+**									
*Reseda lutea*	1			**+**									
*Rubia peregrina*	7		**+**				**+**				**+**	**+**	
*Rumex pulcher*	3		**+**								**+**		
*Salvia verbenaca*	1				**+**								
*Scandix pecten-veneris*	1		**+**				**+**				**+**		
*Senecio jacobaea*	1		**+**								**++**		
*Silybum marianum*	7		**+**				**+**				**+**		
*Sisymbrium runcinatum*	1												**+**
*Solanum nigrum*	2		**+**								**+**		
*Sonchus oleraceus*	2		**+**								**+**		
*Staehelina dubia*	1			**+**									
*Stipa parviflora*	2			**+**								**+**	**+**
*Taraxacum officinale*	3				**+**								
*Teucrium botrys*	1			**+**									
*Teucrium capitatum*	2			**+**									**+**
*Teucrium chamaedrys*	1			**+**								**+**	
*Teucrium pseudochamaepitys*	3			**+**	**+**								
*Thapsia villosa*	2			**+**								**+**	**+**
*Thymus vulgaris*	1			**+**									
*Torilis nodosa*	1		**+**								**+**		
*Trifolium campestre*	1										**+**		
*Verbascum sinuatum*	3				**+**								
*Vicia* sp.	2		**+**				**+**				**+**		
*Xanthium strumarium*	1	**+**											
*Zea mays*	2	**+**											
Host range by habitat (HTS2C)		17	47	32	20	7	28	6	3	9	52	12	12
Host range by ecosystem (HTS2C)		88				38				67			

**+** Detected by HTS2C only; empty grey cell detected by RT-PCR only, both **+** and grey cell detected by RT-PCR and HTS2C.

**Table 2 viruses-15-01779-t002:** Co-occurrence simulation tests for virus-sharing host species among the habitats. For any particular species pair, the larger the C-score, the more segregated the pair, with fewer shared host species.

	Population	Observed C-Score	Simulated Mean C-Score	Cohens *d*	Lower CI *d*	Upper CI *d*	Lower *p*-Value	Upper *p*-Value
Sim2	*Host species*							
	Crop	8.00	9.71	0.43	0.47	0.47	0.241	0.574
	Edge	6.67	19.85	1.51	1.66	1.66	0.017	0.869
	Oak	85.00	40.74	3.31	−3.55	−3.30	0.999	0.001
	Wasteland	50.00	25.17	2.94	−3.20	−2.93	0.999	0.001
Sim4	*Host species*							
	Crop	8.00	7.92	0.02	−0.03	−0.01	0.409	0.389
	Edge	6.67	16.59	1.17	1.29	1.29	0.039	0.765
	Oak	85.00	36.39	3.74	−4.00	−3.72	1.000	0.000
	Wasteland	50.00	22.49	3.31	−3.64	−3.30	0.999	0.001
Sim2	*HTS Libraries*							
	Crop	49.67	29.48	2.05	−2.20	−2.04	0.973	0.021
	Edge	111.00	265.00	2.42	2.66	2.66	0.004	0.996
	Oak	101.33	44.30	3.65	−4.01	−3.63	1.000	0.000
	Wasteland	63.33	28.75	3.42	−3.74	−3.40	1.000	0.000
Sim4	*HTS Libraries*							
	Crop	49.67	25.35	2.53	−2.69	−2.51	0.991	0.007
	Edge	111.00	207.33	1.60	1.76	1.76	0.044	0.955
	Oak	101.33	39.81	4.05	−4.47	−4.03	1.000	0.000
	Wasteland	63.33	26.02	3.74	−4.00	−3.72	1.000	0.000

**Table 3 viruses-15-01779-t003:** Nucleotide diversity (*π*) of CMV across habitats.

Segment	Crop	Edge	Oak	Wasteland	Mean ± SE *π*
RNA1	0.007	0.005	0.006	0.008	0.007 ± 0.001
RNA2	0.007	0.006	0.004	0.007	0.006 ± 0.001
RNA3	0.014	0.005	0.002	0.011	0.008 ± 0.005
Mean ± SE *π*	0.009 ± 0.004	0.005 ± 0.001	0.004 ± 0.002	0.009 ± 0.002	

**Table 4 viruses-15-01779-t004:** Bayesian Tip-association Significance tests (BaTS) of CMV populations at habitat and site levels. Significant results are given in bold text. The ‘Anthropic effect’ grouped Edge and Crop to compare with Oak and Wasteland (i.e., non-anthropic habitats).

	Statistic	Obs. Mean	Obs. L. 95% CI	Obs. U. 95% CI	Null Mean	Null L. 95% CI	Null U. 95% CI	*p*-Value
CMV RNA1								
Habitat effect	AI	0.97	0.56	1.38	2.46	2.03	2.85	0.000
	PS	9.40	8.00	10.00	19.23	17.48	20.69	0.000
Anthropic effect	AI	0.36	0.26	0.60	1.09	0.76	1.45	0.000
	PS	3.00	3.00	3.00	7.39	5.91	8.39	0.000
CMV RNA2								
Habitat effect	AI	0.92	0.38	1.46	2.38	2.10	2.58	0.000
	PS	9.05	9.00	9.00	20.34	18.38	21.61	0.000
Anthropic effect	AI	0.44	0.06	0.77	1.18	0.93	1.38	0.000
	PS	4.01	4.00	4.00	8.74	7.40	9.90	0.000
CMV RNA3								
Habitat effect	AI	1.17	0.71	1.62	2.54	2.08	2.98	0.000
	PS	9.43	8.00	10.00	19.56	17.83	21.16	0.000
Anthropic effect	AI	0.58	0.22	0.91	1.21	0.82	1.54	0.010
	PS	4.36	4.00	5.00	8.77	7.08	10.00	0.000

## Data Availability

The data supporting reported results will be provided by the authors on request.

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
