# Peer review of "Ecological Strategies for Resource Use by Three Bromoviruses in Anthropic and Wild Plant Communities"

_viruses, 2023, doi:10.3390/v15081779_

Round 1

Reviewer 1 Report

The proposed paper from Babalola et al aims to show the differences in ecological strategies, through their host ranges, of three different viruses from Bromoviridae family in four anthropic and non-anthropic environments. The host range is estimated by HTS detection only but results were validated with RT-PCR on a large number of randomly chosen libraries. Genetic diversity along the three viral genomes and population genetic structure for the CMV were also assessed.

The methods and the number of samples seem adapted to answer the questions posed in the introduction. The first two parts of the Materials and Methods could however be improved for an easier understanding of the sequenced samples. Details are provided in a cited previous paper (McLeish et al, 2022), but more information in the section could help the reader not to get lost through the different levels of sampling (collection, sites, habitats…).

The results obtained are numerous and very interesting. The discussion resumes the main results but the integration in a scientific context could be improved, in particular for parts concerning the genetic diversity and the population structure, where no bibliographical reference is quoted. Be careful not to confuse co-occurence of the viruses in a same species with co-infection in a same plant. The pooled results do not allow conclusions to be drawn at the individual level, and the relevant part of the discussion (line 446 to 456) should recall this fact.

Here are some minor suggestions to improve the paper:

-          line 151: change 2C to HTS2C to be consistent in the whole paper

-          line 295: change Fig.ure to Figure.

-          whole document (eg. Lines 279,296, 423): since results are not based on individual sequencing but on pooled samples, terms like co-infection or multiple infection should be avoid. The term occurrence, already used in the paper, is more appropriate.

-          Table 1: (1) please double check the table ; 17 positive detection in the CR/CMV line (only 16 in the table) and 87 in the CMV host range by ecosystem (88 in the table); (2) What does “++” means in PZSV ? ; (3) a header on each page or different cells background by virus would improve the reading of the table

Author Response

Reviewer 1:

The proposed paper from Babalola et al aims to show the differences in ecological strategies, through their host ranges, of three different viruses from Bromoviridae family in four anthropic and non-anthropic environments. The host range is estimated by HTS detection only but results were validated with RT-PCR on a large number of randomly chosen libraries. Genetic diversity along the three viral genomes and population genetic structure for the CMV were also assessed.

The methods and the number of samples seem adapted to answer the questions posed in the introduction. The first two parts of the Materials and Methods could however be improved for an easier understanding of the sequenced samples. Details are provided in a cited previous paper (McLeish et al, 2022), but more information in the section could help the reader not to get lost through the different levels of sampling (collection, sites, habitats…).

Response: We have now added a detailed explanation of the sampling levels and the sample sizes used for the high throughput approach, which is now provided in the Methods section of the new version.

The results obtained are numerous and very interesting. The discussion resumes the main results but the integration in a scientific context could be improved, in particular for parts concerning the genetic diversity and the population structure, where no bibliographical reference is quoted. Be careful not to confuse co-occurence of the viruses in a same species with co-infection in a same plant. The pooled results do not allow conclusions to be drawn at the individual level, and the relevant part of the discussion (line 446 to 456) should recall this fact.

Response: The content that discusses the genetic diversity and population structure has now been set in a broader context supported with additional references. We agree with the reviewer and had made every attempt not conflate the use co-occurrence with co-infection. The lines indicated by the reviewer do not seem to be relevant to co-infection at all. “[C]oinfection” was only used once in the text (previously line 423). This instance has been omitted in the revised version.

Here are some minor suggestions to improve the paper:

-          line 151: change 2C to HTS2C to be consistent in the whole paper

Response: All instances of ‘2C’ have been amended to ‘HTS2C’.

-          line 295: change Fig.ure to Figure.

Response: The typo has now been corrected.

-          whole document (eg. Lines 279,296, 423): since results are not based on individual sequencing but on pooled samples, terms like co-infection or multiple infection should be avoid. The term occurrence, already used in the paper, is more appropriate.

Response: The authors agree and this instance has been omitted.

-          Table 1: (1) please double check the table ; 17 positive detection in the CR/CMV line (only 16 in the table) and 87 in the CMV host range by ecosystem (88 in the table); (2) What does “++” means in PZSV ? ; (3) a header on each page or different cells background by virus would improve the reading of the table

Response: The inconsistency has been corrected. The header format for the Table will be at the discretion of the copy editor.

Reviewer 2 Report

Authors Bisola Babalola and co-workers presented a manuscript entitled "Ecological strategies for resource use by three bromoviruses in anthropic and wild plant communities".

They studied the ecological strategies and interrelationships of three bromoviruses infecting different host plant species in four habitats of an ecosystem in central Spain. They used HTS analysis of pooled samples to study population genomics and genetic variability of the viruses studied.

The study provides a complex view of the relationships of viruses in their environment and merits publication in Viruses.

Suggestions for improving the manuscript:

Chaotic page numbering

Page 3, lines 114-117. Sentence about the results of the work should be better placed in the conclusion or abstract.

Page 25, lines 601-603: numbering of references - need to delete previous version.

Author Response

Reviewer 2:

Authors Bisola Babalola and co-workers presented a manuscript entitled "Ecological strategies for resource use by three bromoviruses in anthropic and wild plant communities". They studied the ecological strategies and interrelationships of three bromoviruses infecting different host plant species in four habitats of an ecosystem in central Spain. They used HTS analysis of pooled samples to study population genomics and genetic variability of the viruses studied.

The study provides a complex view of the relationships of viruses in their environment and merits publication in Viruses.

Suggestions for improving the manuscript:

Chaotic page numbering

Response: The page numbering is the format embedded in the template provided by the journal. As such, we will leave the copy editor to produce the appropriate numbering in the print version

Page 3, lines 114-117. Sentence about the results of the work should be better placed in the conclusion or abstract.

Response: The authors appreciate that a summary at the end of the Introduction section is not the practice of many journals, but has been adopted as a useful device by many others. We elect to retain the text as it provides an indication to the way the following content is developed.

Page 25, lines 601-603: numbering of references - need to delete previous version.

Response: The extraneous numbering has been deleted.